# Cell Membrane-Derived Nanovehicles for Targeted Therapy of Ischemic Stroke: From Construction to Application

**DOI:** 10.3390/pharmaceutics16010006

**Published:** 2023-12-19

**Authors:** Cui Hao, Ma Sha, Yang Ye, Chengxiao Wang

**Affiliations:** 1School of Life Science and Technology, Kunming University of Science and Technology, Kunming 650500, China; kustch@163.com (H.C.); 18987359334@163.com (S.M.); yangye@kust.edu.cn (Y.Y.); 2Key Laboratory of Sustainable Utilization of Panax Notoginseng Resources of Yunnan Province, Kunming 650500, China

**Keywords:** ischemic stroke, biomimetic systems, cell membranes, formulation, review

## Abstract

Ischemic stroke (IS) is a prevalent form of stroke and a leading cause of mortality and disability. Recently, cell membrane-derived nanovehicles (CMNVs) derived from erythrocytes, thrombocytes, neutrophils, macrophages, neural stem cells, and cancer cells have shown great promise as drug delivery systems for IS treatment. By precisely controlling drug release rates and targeting specific sites in the brain, CMNVs enable the reduction in drug dosage and minimization of side effects, thus significantly enhancing therapeutic strategies and approaches for IS. While there are some reviews regarding the applications of CMNVs in the treatment of IS, there has been limited attention given to important aspects such as carrier construction, structural design, and functional modification. Therefore, this review aims to address these key issues in CMNVs preparation, structural composition, modification, and other relevant aspects, with a specific focus on targeted therapy for IS. Finally, the challenges and prospects in this field are discussed.

## 1. Introduction

Ischemic stroke (IS), characterized by the obstruction of cerebral blood vessels, results in reduced blood flow, insufficient oxygen and nutrient supply, and subsequent neuronal death [1], and is the most prevalent type of stroke and a leading cause of mortality and disability [2,3]. Current treatment approaches for IS include thrombolytic therapy, mechanical thrombectomy, pharmacological interventions, and rehabilitation [4,5]. However, most of these treatments have a limited therapeutic window and are only effective for a small number of patients. Neuroprotection and neurorecovery interventions are crucial for salvaging the ischemic penumbra in patients in whom the optimal treatment window has been missed. Unfortunately, the clinical application of neuroprotective drugs is hindered by their poor bioavailability due to their limited diffusion through the blood–brain barrier [6].

Nano delivery systems are based on nanotechnology and offer distinct advantages in the treatment of IS [7]. These systems enable precise delivery of drugs to the brain by controlling drug release rates and targeting specific sites, thereby reducing drug dosage and minimizing side effects. Furthermore, nanoparticles can cross the blood–brain barrier and reach cerebral tissues, allowing for direct drug action in the affected area and enhancing therapeutic efficacy [8,9,10,11]. Various nanoplatforms, including liposomes, micelles, polymeric nanoparticles, nanogels, and inorganic nanomaterials, have shown promising potential in the treatment of IS [10,12,13]. However, the application of these nanocarriers is limited by factors such as poor targeting, drug leakage, inadequate stability and biocompatibility, as well as a short blood circulation time [14]. These factors contribute to reduced drug delivery efficiency and poor therapeutic effectiveness [15]. Therefore, there is a significant value in exploring the development of novel nano delivery systems that deviate from traditional approaches.

Among nanoplatforms, cell membrane-derived nanovehicles (CMNVs) have emerged as highly promising drug delivery systems [16,17]. CMNVs can be categorized into two main types, namely, rigid nanoparticles (NPs) with a core-shell structure (i.e., cell vesicles are attached to synthesized NPs) and nanovesicles (i.e., therapeutic molecules are encapsulated within cell vesicles) [18]. The cell membrane coating strategy is an excellent biomimetic tool utilized in various therapies, particularly in targeted chemotherapy. Uncoated drug-loaded nanoparticles have drawbacks, such as being easily recognized and attacked by the reticuloendothelial system, leading to their rapid elimination from the body. However, CMNVs can overcome these issues [19]. They possess the unique ability to mimic endogenous cells in the body, thereby exhibiting high biocompatibility. Specifically, CMNVs have demonstrated unprecedented potential in the treatment of brain diseases. Utilizing endogenous substances of natural origin, CMNVs achieve efficient passage across the BBB by simulating specific pathological conditions associated with brain diseases [20]. Moreover, CMNVs can specifically target organelles in different pathological environments after crossing the BBB, thereby exerting effective multi-targeting effects in treating specific diseases [21].

Most importantly, CMNVs serve as nano-biomimetic systems by possessing essential proteins on their outer cell membrane, allowing for cell–cell interactions and contributing to crucial capabilities like immune evasion, BBB crossing, and targeted delivery to brain lesions. For instance, proteins such as CD47, CD59, C8-binding protein, and complement factors are widely distributed on the red blood cell membrane and effectively inhibit macrophage phagocytosis [22,23]. Consequently, nanoparticles modified with red blood cell membranes can evade immune system attacks and circulate for longer periods. Another example is CD138, a polysaccharide protein-1 present on the membrane of 4T1 cancer cells, which binds to endothelial cell adhesion molecule-1 (CD31) found on platelets, endothelial cells, and leukocytes. This binding promotes adhesion and facilitates migration across the BBB, facilitating BBB crossing delivery [24,25,26]. Thrombocytes, with their surface integrins (such as αIIb, α2, α5, α6, β1, β3) and other transmembrane proteins (like GPIbα, GPIV, GPV, GPVI, GPIX, and CLEC-2), confer the ability to target blood clots [27]. Furthermore, leukocyte surface proteins such as CD11, CD18, and CD44 specifically recognize and bind to intercellular adhesion molecule-1 (ICAM-1) and P-selectin on endothelial cells, displaying high targeting efficiency in various inflammatory diseases [28].

The application of CMNVs in the treatment of IS has recently been reviewed. Li et al. and Liao et al. provided overviews of the applications of CMNVs in various brain diseases, including gliomas, IS, Parkinson’s disease, and Alzheimer’s disease [29]. Liu et al. specifically focused on the progress of CMNVs in IS treatment. Notably, many review articles on this topic begin with the source of cell membranes, and literature reviews are usually based on the classification of cell membranes [21]. This approach, guided by the functional differences of cell membranes (such as erythrocytes, thrombocytes, and macrophages), summarizes the research progress of CMNVs constructed with different cell membranes in the context of IS. This method facilitates readers’ understanding of the functions of the cell membranes. However, certain critical aspects of CMNVs, such as carrier construction, structural design, and functional modification, have not received sufficient attention.

Therefore, in this review, we focused on targeted therapy for IS by addressing the common issues in CMNVs preparation, structural composition, modification, and other relevant aspects. We deconstructed and condensed the literature, and discussed key factors in the design and construction process of CMNVs to establish their intrinsic connection with IS treatment. Subsequently, we summarized the progress of CMNVs in targeted therapy for IS, starting from the pathological features of IS. Finally, we discussed the technical difficulties and challenges in the application of CMNVs for IS treatment and highlighted future prospects. We hope that this review will offer a novel research perspective on the design, construction, and application of CMNVs.

## 2. Characterization of Ischemic Stroke

Stroke occurs when there is insufficient blood supply to the brain due to the rupture or blockage of cerebral blood vessels. It is classified into two main types, namely, IS and hemorrhagic stroke, where IS is the more prevalent form. The primary pathogenesis of IS involves the obstruction of cerebral blood vessels by thrombus formation or atherosclerosis [30,31]. Hemorrhagic stroke represents intracerebral bleeding that occurs when cerebral blood vessels rupture due to arterial aneurysms or hypertension [32,33]. In the case of IS, brain cells die due to insufficient oxygen and nutrient supply and accumulation of metabolic waste products. In hemorrhagic stroke, cerebral bleeding causes mechanical damage to surrounding tissues, which further exacerbates ischemic conditions [34,35,36].

Cerebral infarction is a common subtype that results from the obstruction of cerebral blood vessels and consequent inadequate blood supply to the brain [37,38]. The main pathogenesis of cerebral infarction involves atherosclerosis, thrombus formation, and embolism. These mechanisms interact to impede cerebral blood flow, ultimately resulting in ischemia and hypoxia in the brain and causing IS [39].

Cerebral ischemia-reperfusion injury refers to a series of pathological and physiological reactions, including cellular damage, that occur after the restoration of blood supply following cerebral ischemia [40,41]. This type of injury commonly occurs in situations such as stroke, cardiac arrest, and cardiac surgery, where cerebral blood supply is temporarily halted and subsequently reintroduced, which leads to more severe damage in the brain tissue. The pathogenesis of cerebral ischemia-reperfusion injury involves two stages, namely, the ischemic phase injury and the reperfusion phase injury [42]. Cellular energy depletion and glutamate release during ischemia, along with the generation of free radicals, activation of inflammatory responses, the release of toxic substances, and disruption of the blood–brain barrier during reperfusion, together cause irreversible damage to brain cells.

## 3. Characterization of the Cell Membrane

In the human body, different cell types play crucial roles in various physiological functions. These functions include long-term circulation in the bloodstream, migration to specific regions of the body, and penetration of physiological barriers. The selection of specific cell types is essential for delivering drugs while preserving the cellular structure and function.

### 3.1. Erythrocytes

Erythrocytes are biconcave-shaped blood cells that lack a cell nucleus and organelles. They are a vital component of blood and have a longer lifespan. These cells possess a high surface area to volume ratio, which facilitates efficient supply and transport [43,44]. Importantly, carriers based on erythrocytes have immense engineering potential and can be utilized for the delivery of various drugs [45]. Earlier cell membrane-based biomimetic delivery systems employed the membrane of erythrocytes. Recently, the membrane of erythrocytes has attracted interest as a biomimetic brain delivery system due to erythrocytes’ availability, prolonged circulation time, uniform size, and shape [44]. In comparison to other cell membranes, the modification of nanocarriers on red blood cell membranes primarily results in long circulation and stealth properties. However, to attain brain-targeted characteristics, additional functional modifications are necessary [46].

### 3.2. Thrombocytes

Thrombocytes, akin to cell-like structures, lack a nucleus and possess a disc-shaped morphology. They are less numerous, smaller, and shorter-living than erythrocytes [47], and actively partake in immunological, inflammatory, and thrombotic processes [48]. Their aggregation is crucial for hemostasis. In recent years, thrombocytes have attracted significant attention in the field of drug delivery [49] due to their strong storage and transportation capabilities [50] and natural targeting and adhesive properties [51]. For brain-targeted delivery systems, thrombocyte membrane possesses inherent targeting abilities and can specifically target damaged blood vessels, which makes it particularly suitable for the treatment of IS [52]. Recently, a novel nanoparticle coated with platelet membranes has been developed. This biomimetic system effectively preserves the functionality of platelets while allowing the nanoparticles to avoid immune system detection. The CMNVs can enter neutrophils via endocytosis and exploit the inflammatory chemotaxis of neutrophils to traverse the blood–brain barrier (BBB) [53].

### 3.3. Leukocytes

Leukocytes play a vital role in maintaining the body’s immune defense. They include various subtypes such as neutrophils, eosinophils, basophils, monocytes, and lymphocytes. Leukocytes possess adhesive capabilities, which allow them to interact with and adhere to specific tissues or cells in the body [54]. Moreover, leukocytes exhibit migratory and chemotactic abilities under different pathological conditions, thereby enabling drug delivery to diseased areas [55]. Notably, neutrophil-homing receptors such as Mac-1 and LFA-1 can guide drug delivery systems to the damaged brain tissue by interacting with endothelial cells in inflamed microvessels. Inspired by the interaction between neutrophils and brain endothelial cells in the pathogenesis of ischemic stroke, Dong et al. [56] reported a drug delivery system constructed from neutrophil nanovesicles that can specifically target inflamed brain endothelium in an ischemic stroke mouse model. This system enhances the resolution of inflammation during ischemic stroke therapy. In addition, recent studies have utilized surface modification mechanisms based on neutrophil-associated ligands, offering promising prospects for targeted treatment of brain diseases [57,58].

### 3.4. Natural Killer Cells

Natural killer (NK) cells are a type of immune effector cells characterized by the presence of numerous granules in the cytoplasm. NK cells are primarily found in the peripheral blood and spleen. In contrast to specific or pan-specific antigen recognition receptors, NK cells do not express these receptors. Instead, they express a variety of regulatory receptors associated with activation and inhibition. These regulatory receptors enable NK cells to discern between “self” and “non-self” components of the body, allowing them to selectively kill target cells such as virus-infected cells or tumor cells [59,60]. Recent clinical studies have shown that immune cells can bypass the blood–brain barrier (BBB) and perform immune surveillance in the CNS by utilizing specific membrane proteins as adaptors. Deng et al. [61] reported a bio-mimic nanoparticle by encapsulating an NK cell membrane on a nanorobot, which is capable of autonomously crossing the BBB by disrupting the tight junction (TJ) structure, and specifically accumulating in brain tumor tissues within the complex brain environment.

### 3.5. Macrophages

Macrophages are a specialized type of leukocytes in the immune system, the primary function of which is to phagocytose and digest pathogens such as bacteria, viruses, fungi, and parasites to maintain the normal functioning of the immune system. During ischemia and reperfusion processes, the integrity of the blood–brain barrier is partially compromised, which leads to the recruitment and infiltration of macrophages into the ischemic brain within a few hours. Consequently, enveloping NPs with macrophage membranes may be a promising strategy for precise delivery to the site of ischemic injury [62,63]. Li et al. developed manganese dioxide (MnO_2_) nanospheres coated with macrophage membranes, enabling them to actively accumulate in the damaged brain through recognition mediated by macrophage-membrane proteins and cell adhesion molecules that are overexpressed on the damaged vascular endothelium [64]. Recently, macrophage membranes (MMs) with active targeting inflammation function were used to encapsulate quantum dots, which enabled effective targeting of the brain and enhanced the ability of substances to cross the blood–brain barrier (BBB) [65].

### 3.6. Neural Stem Cells

Stem cells are a versatile cell type with self-renewal capacity, multipotent differentiation potential, and low immunogenicity. Neural stem cells (NSCs) have been explored for the treatment of IS due to their significant tropism towards ischemic brain regions. Biomimetic drug delivery systems based on NSCs can achieve effective targeted delivery [66]. Wu et al. [67] utilized the bio-responsiveness of stem cell membranes to achieve targeted therapy for inflamed BBB. They prepared hybrid bio-responsive vesicles (NSC-Lipo) by recombining NSC membranes with simple liposomes. The expression of VLA-4 on NSC membranes endowed NSC-Lipo with specific recognition and binding capability to VCAM-1 on injured brain microvessel endothelial cells (BMECs), enabling NSC-Lipo to target and be endocytosed into the lesion area rapidly.

### 3.7. Cancer Cells

Tumor cells are an abnormal cell population formed by the malignant mutation of normal cells. The interaction between the membrane molecules of cancer cells and the receptors of endothelial cells is crucial for the attachment of cancer cells to brain endothelial cells and subsequent trans-BBB migration. This cell–cell interaction is multivalent and involves multiple substances, which contributes to the superior blood–brain barrier penetration efficacy of brain metastatic cancer cells compared to most developed brain-targeted systems [24,25,26]. In a reported work, a polymer nanoparticle coated with the membrane of brain metastatic breast cancer cells (MDA-MB-831) (CCNP) was synthesized, which demonstrated effective penetration and retention ability in the brain [68]. Recently, a nanocarrier utilizing the Trojan horse strategy has been developed by integrating the cell membrane of brain-targeted cancer cells (MDA-MB-231/breast cancer cell) with a polymeric drug depot. This strategy takes advantage of the camouflage effect of the cell membrane to facilitate penetration through the blood–brain barrier (BBB) [69].

### 3.8. Extraction and Purification of Cell Membranes

Cell membranes comprise an asymmetric phospholipid bilayer and embedded functional surface proteins [70]. The functional surface proteins play a crucial role in cell recognition and interaction, targeted delivery, and immune regulation. The loss of these proteins leads to the termination of essential cellular functions [71]. Considering that the functional surface proteins may be inactivated under various conditions, the extraction of cell membranes requires careful consideration. The specific process of extraction depends on the type of cells being targeted. For eukaryotic cells such as leukocytes, cancer cells, and stem cells, the extraction process is more complex and involves cell separation, purification, culture, and passaging. Common methods for membrane extraction include hypotonic methods, freeze-thaw methods, ultrasound methods, and chemical solvent methods [44,72]. While these methods may differ in certain aspects, their core goal is to break down cells physically or chemically while preserving the integrity of the cell membrane. This enables the effective separation of the cell contents from the cell membrane, followed by the purification and enrichment of the desired membrane proteins. In conclusion, there are various methods for extracting cell membranes, and all of them have their own principles and steps. When selecting a method, it is important to take into account experimental requirements, cell types, required components, and subsequent experimental or analytical needs. Different methods have their own advantages and limitations, and the most suitable method should be chosen based on specific circumstances.

## 4. Cell Membrane-Derived Nanovehicles with a Core-Shell Structure

NPs with a core-shell structure are widely used as biomimetic delivery systems. These systems are constructed using a bottom-up approach, where synthetic NPs are modified with cell vesicles. In this unique structure, the cell membrane functions as the outer shell, providing essential features such as immune evasion, high permeability, and targeted delivery. The nanocore acts as a drug carrier and offers functionalities such as sustained release, controlled release, and stimulus-responsive release. The schematic diagram of the production process for CMNVs with a core-shell structure is shown in Figure 1, while the major information about these CMNVs is given in Table 1.

### 4.1. Nanocore

When developing biomimetic delivery systems for targeting the central nervous system, various types of nanocores are utilized. These include polymer NPs, metal NPs, nano enzymes, solid lipid NPs (SLNs), and other functionalized nanocores.

#### 4.1.1. Polymer Nanoparticles

Polymer NPs are nano-sized particles that comprise polymer materials, which are primarily prepared using methods such as solvent precipitation, emulsification, and self-assembly. The shape, size, and surface properties of the polymer NPs can be precisely adjusted by careful control of the solvent, temperature, and additives. By encapsulating drugs within the particles, these NPs can be used as effective drug delivery systems, and targeted release can be achieved. This approach has the potential to enhance the therapeutic effect of drugs while minimizing the risk of side effects.

##### PLGA NPs

PLGA NPs are widely used in cell membrane mimetic delivery systems. PLGA is a copolymer that comprises lactic acid and glycolic acid, possessing excellent biocompatibility and biodegradability. Thus, PLGA is an ideal carrier for drug delivery and biomedical applications. These NPs have the ability to encapsulate a variety of drugs, including water-soluble and lipid-soluble substances. Encapsulation is achieved through dissolution, microemulsion, or solid dispersion, resulting in the formation of stable nanosystems. In the context of cell membrane mimetic delivery systems, PLGA can effectively load drugs such as curcumin, rapamycin [73], atorvastatin calcium [74], and other medications [75] for targeted delivery to the central nervous system. Moreover, empty PLGA NPs can serve as nanocores, providing rigidity and mechanical strength to CMNVs and improving their physical stability [76,77].

##### Human Serum Albumin

Human serum albumin (HSA), derived from human plasma, is recognized for its high compatibility with the human body, primarily due to its composition closely resembling that of human plasma proteins [95]. Consequently, HSA offers a superior safety profile and markedly reduces the risk of eliciting an immune response, making it exceptionally suitable for various medical applications. Currently, there are several primary methods for preparing HAS NPs, including desolvation, self-assembly, thermal gelation, spray drying, and emulsification techniques [96]. These NPs can encapsulate drugs, thereby safeguarding them from degradation and extending their circulation time in the body. In a recent study [78], human serum albumin (HAS) has been employed to formulate albumin NPs loaded with curcumin. Subsequently, these NPs were coated with erythrocyte membranes to facilitate the targeted treatment of Alzheimer’s disease.

##### Polydopamine NPs

Polydopamine NPs (PDAs) are nano-sized polymers formed through oxidative polymerization of dopamine molecules. Drug molecules can be immobilized onto the surface of PDAs using physical adsorption or covalent bonding, allowing for the controlled release of the drugs. Moreover, PDAs exhibit favorable biocompatibility and biodegradability, thereby facilitating metabolism and elimination from the body. PDA surfaces possess a multitude of functional groups that can be readily modified, thus granting them significant potential in the functionalization and customization of nanomaterials. In the realm of biomimetic delivery systems, surface modification of PDAs with Zn^2+^ particles, which have low toxicity, has been employed to enhance their capacity for loading the A151 protein [79].

##### Dextran NPs

Dextran is a polysaccharide comprising glucose molecules connected via α-1,6-glycosidic bonds. The preparation of dextran NPs can be achieved through the processes of nanoprecipitation and self-assembly. Previous studies have successfully loaded bortezomib into functionalized dextran nanocores with a size of approximately 100 nm. These nanocores were subsequently encapsulated in thrombocyte membranes to produce the so-called CMNVs [97]. Another approach involves modifying dextran to form amphiphilic polymers, which then self-assemble into NPs [80,81]. Dextran-g-PBMEO (poly (benzyl methacrylate)-grafted dextran) can self-assemble into nanostructures, serving as biomimetic nanocores that enhance drug loading through π–π stacking interactions with the drug [82].

#### 4.1.2. Metal Dioxide Nanoparticles

Manganese dioxide (MnO_2_) NPs exhibit efficient catalytic activity in the decomposition of hydrogen peroxide (H_2_O_2_) to produce oxygen (O_2_). This characteristic makes MnO_2_ NPs suitable for use as oxygen delivery systems and for the treatment of hypoxia-related diseases. In a recent study [64], MnO_2_ NPs have been used as nanocores to adsorb the drug FTY. The surface of the NPs has then been coated with macrophage membranes, creating a biomimetic delivery system that mimics macrophages for immune therapy at the site of cerebral ischemia. Iron oxide (Fe_2_O_3_) NPs possess magnetic properties, which facilitate their guidance to specific regions through the use of an external magnetic field, thereby enhancing treatment precision and effectiveness. Li et al. developed a biomimetic magnetic nanoparticle system by encapsulating γ-Fe_2_O_3_ NPs loaded with L-arginine within thrombocyte membranes [83]. This system was employed for efficient delivery at thrombotic sites.

#### 4.1.3. Nanoenzymes

Nanoenzymes are nanomaterials that possess enzyme-catalyzed properties and can be easily prepared, controlled in size, and adjustable in function. These nanoenzymes have attracted considerable attention and have made significant advancements in the field of disease diagnosis and treatment, including biological detection, anti-inflammatory therapy, anti-oxidative damage prevention, and cancer treatment [98]. Feng et al. synthesized nanoenzymes with anti-inflammatory and antioxidant activities using polyvinylpyrrolidone (PVP) and potassium ferrocyanide and coated them with neutrophil membranes [84]. Fu et al. developed a targeted nanoenzyme system by functionalizing cerium oxide-zoledronic acid NPs as nanocores and coating them with thrombocyte membranes, enabling site-specific delivery at thrombotic sites [85]. 

#### 4.1.4. Solid Lipid Nanoparticles

Solid lipid nanoparticles (SLNs) are biocompatible lipid-based nanoscale particles that remain solid at room temperature and are further stabilized with surfactants. They are formed using techniques such as high-pressure homogenization or sonication. SLNs serve as nano-carriers in drug delivery systems. One of the key advantages of SLNs is their ability to encapsulate both hydrophilic and hydrophobic drugs, allowing for effective delivery of a wide range of therapeutic agents [99]. Han et al. utilized a solvent injection method and glyceryl monostearate as the raw material to produce SLNs, which were loaded with genistein [86]. These SLNs were coated with macrophage membranes to facilitate targeted therapy for Alzheimer’s disease.

#### 4.1.5. Other Nanocarriers

Modifying polymer materials allows for the production of functionalized nanocarriers that are responsive to stimuli and enable controlled drug delivery. In biomimetic delivery systems, materials such as PEI, heparin, and even natural polysaccharides can be modified and utilized to prepare nanocarriers. Guo et al. conducted a study in which they utilized PEI to incorporate pH-sensitive phenylboronic acid and construct a thrombus microenvironment-responsive nanocore. This nanocore was subsequently coated with thrombocyte membranes to enhance its targeting efficacy towards sites of thrombosis [87]. Heparin, a highly effective anticoagulant drug, can undergo self-assembly with thiolated polylysine (PLL-SH) through disulfide bonding (S-S) to form a nanocore [88]. Su et al. utilized biologically active Angelica polysaccharide as the hydrophilic end, and linked ethyl ferulate to the ROS-responsive oxalate bond, forming ROS-responsive nanocores, which were then camouflaged with macrophage membrane. This approach achieved targeted delivery of the biologically active macromolecules to brain thrombosis [89].

### 4.2. Fusion of Membrane Vesicles and Nanoparticles

There are several methods available for coating NPs with cell membranes, including co-extrusion [90], sonication [100], and microfluidic electroporation [101]. Co-extrusion involves applying physical pressure using an extruder to mix purified cell membranes with NPs, which serve as the “core”. The resulting mixture is then co-extruded through a porous membrane [102]. This technique is commonly utilized in the preparation of liposomes, as the mechanical force from extrusion can disrupt the membrane structure and allow the cell membrane to reassemble around the NPs, thereby creating a core-shell structure.

Sonication employs the disruptive force generated by ultrasound energy to spontaneously form core-shell nanostructures comprising NPs and cell membranes. NPs prepared using this method not only yield consistent results comparable to those obtained through physical extrusion, but also offer the advantage of minimal material loss. The membrane coating mechanism is generally attributed to the semi-stable nature of vesicles derived from the core of NPs and the asymmetric charge distribution of biological membranes, which promote the formation of a core-shell structure with a membrane orientation from the inside out (right-side out) [103].

In microfluidic electroporation, in conjunction with a rapid mixing microfluidic system, magnetic NPs have been successfully coated with erythrocyte membranes. The device comprises a Y-shaped merging channel, an S-shaped mixing channel, an electroporation zone, and an outlet. By precisely adjusting the pulse voltage, duration, and flow rate, erythrocyte membranes are effectively coated onto the NPs with exceptional stability [101].

## 5. Cell Membrane Nanovesicles

Cell membrane nanovesicles are a type of flexible NPs with a hollow structure. They are similar to liposomes but distinct from biomimetic core-shell NPs. These nanovesicles are prepared using a top-down approach, where therapeutic molecules are encapsulated within the vesicles. Classic methods for producing cell membrane nanovesicles include ultrasonic disruption and thin-film extrusion. For instance, He et al. incubated macrophages with a high concentration of curcumin, allowing curcumin to be engulfed by the macrophages through phagocytosis. The resulting extracellular vesicles derived from macrophages, loaded with curcumin, were collected and utilized for the treatment of stroke [104]. Similarly, Vankayala et al. utilized a low-osmotic centrifugation method to prepare functionalized erythrocyte membrane vesicles, loaded with ICG and tPA, for the integrated diagnosis and treatment of thrombi [105]. Another study by Shen et al. involved the continuous extrusion of macrophages through filters with gradually decreasing pore sizes to obtain cell membrane vesicles loaded with curcumin, facilitating enhanced neuronal uptake of the drug for the treatment of Parkinson’s disease [106].

Additionally, traditional methods employed in liposome preparation can be utilized to generate cell membrane vesicles by co-formulating cell membranes with phospholipids, cholesterol, and other lipids. The inclusion of cholesterol significantly improves the stability and dispersion of these lipid-based nanosystems. For example, Wu et al. isolated cell membranes from mesenchymal stem cells and prepared hybrid vesicles through thin-film dispersion. These lipid hybrid vesicles, comprising stem cell membranes and phospholipids, were loaded with curcumin for the treatment of IS. These vesicles exhibit a comparable particle size (70~80 nm) to conventional liposomes [107]. In another study, Zhang et al. isolated essential membrane proteins from macrophages and formulated a brain-targeted biomimetic delivery system by co-formulating them with phospholipids and long intergenic noncoding RNA (lincRNA-EPS) using the thin-film dispersion method [108]. Moreover, for the treatment of Parkinson’s disease, Liu et al. successfully fused natural killer (NK) cell membranes with curcumin-loaded liposomes using thawing and extrusion techniques, which resulted in the formation of an NK cell-membrane biomimetic nanocomplex [109]. The schematic diagram of the production process for cell membrane nanovesicles is shown in Figure 2, while the major information about these nanovesicles is shown in Table 2.

## 6. Drug-Loading Modes

In CMNVs, two common drug-loading modes are prevalent, namely, loading drugs into the core-shell structure of NPs or loading drugs into the internal cavity of cell membrane nanovesicles. Polymer materials can be used to form drug-loaded NPs through processes such as nanoprecipitation or self-assembly in micelles, allowing for the simultaneous incorporation of drugs. It is important to consider the dissolution characteristics of drugs in the reaction medium during this process. Enhancing the interaction between polymers and drug molecules can improve the drug-loading capacity. For example, modified cyclodextrin NPs [82] can increase their drug-loading capacity through π–π stacking with kaempferol.

Drug-polymer conjugates, formed through covalent linkage between drugs and polymers, can also be utilized to produce polymerdrug complexes and subsequent NPs, enhancing drug loading. Ma et al. copolymerized dexamethasone with reactive monomers to design a dexamethasone (Pred) prodrug copolymer [PMPC-P(MEMA-co-PDMA)], which self-assembled into micelles [91]. These micelles could encapsulate erythrocyte membranes to form biomimetic NPs. Zhong et al. used rapamycin as a drug and connected it to PEG through ROS-sensitive groups to synthesize a PEG-OX-RAP prodrug. This amphiphilic polymer may self-assemble into nanocores in an aqueous environment and could be coated with engineered endothelial cell membranes, thereby forming a biomimetic delivery system [92]. Heparin, one of the most effective anticoagulant drugs, can self-assemble with thiolated polylysine (PLL-SH) through disulfide bonding (S-S) to form NPs [88]. In another study [93], doxorubicin (DOX) and succinylated heparin were covalently linked to form amphiphilic DOX-heparin polymers. These polymers may self-assemble into NPs in an aqueous environment and could be coated with thrombocyte membranes.

Drugs with a high affinity for the phospholipid structure of cell membranes can be embedded in the phospholipid bilayer as part of the nanocarrier system. Ginsenosides have a pentacyclic triterpenoid nucleus structure similar to cholesterol. Wang et al. replaced cholesterol molecules with ginsenoside Rg5 in the phospholipid bilayer to prepare biomimetic liposomes, which exhibited good drug-loading capacity and demonstrated good preclinical safety [110]. Furthermore, drugs can be loaded onto the surface of biomimetic delivery systems by modifying the cell membrane to improve drug-adsorption capability. Drugs can be directly linked to membrane proteins through reactions such as acylation, click chemistry, or esterification to achieve stable loading [114,115]. However, the direct conjugation of drugs to the external surface of the membrane structure does not provide adequate protection. Xu et al. connected the thrombolytic drug rt-PA to the modified thrombocyte membrane surface through double-functionalized maleimide, thereby forming a surface-loaded drug biomimetic delivery system for brain targeting [76]. Vankayala et al. modified erythrocyte membrane biomimetic delivery systems and successfully loaded tPA on the surface, which resulted in an enhanced drug half-life and circulation time in the body [105] (Figure 3).

## 7. Surface Modification of Cell Membranes

Cell membranes comprise lipids, polysaccharides, and proteins, and the functionality of biomimetic NPs relies on the proteins on their surface. The surface modification and engineering of biomimetic delivery systems primarily focus on modifying the cell membranes. By modifying and engineering the cell membranes, the stability of the system can be improved, its targeting ability can be enhanced, and new functionalities can be added, ultimately leading to improved therapeutic effects in the diseases of the central nervous system. Cell membrane modification can be broadly categorized into three main approaches, including physical modification, chemical modification, and genetic engineering (Figure 4).

### 7.1. Physical Modification

Physical modification is a commonly used method for cell membrane modification, where target molecules are naturally anchored onto the cell membrane through lipid–lipid interactions [116]. There are three main approaches, namely, inserting functionalized hydrophilic lipid portions into the outer leaflet of the lipid bilayer, inserting exogenous receptors into the cell membrane for specific binding with representative molecules, and fusing cell membranes with other phospholipid components. However, the potential instability of the inserted molecules and their negative effect on the overall stability may limit the effectiveness of this strategy.

The insertion of functional molecules into the phospholipid bilayer is a commonly used physical modification method in the case of core-shell biomimetic NPs. In that process, DSPE-PEG—a functionalized block copolymer—is often employed. DSPE is a phospholipid-like structural unit with a high affinity for cell membranes, and it can spontaneously be inserted into the membrane. Modifying the hydrophilic end of DSPE-PEG can provide additional functionality to biomimetic systems. Targeting peptides, such as stroke homing peptide (SHp) [81], clot-binding peptide CREKA (Cys-Arg-Glu-Lys-Ala), Arg-Gly-Asp (RGD) peptides [75,117], and T807 (AV-1451), have been attached to the hydrophilic end of DSPE-PEG to allow for targeted modification of the cell membrane, improved penetration through the blood–brain barrier, and more precise targeting of brain lesions. Han et al. [86] modified the surface of macrophage membrane biomimetic delivery systems with functional molecules RVG29-PEG-DSPE and TPP-PEG-DSPE for targeted therapy of Parkinson’s disease.

Fusion of cell membrane fragments with phospholipid components is a commonly employed physical modification method for vesicle-like biomimetic NPs. Zhang et al. used a thin-film evaporation method to fuse macrophage membrane fragments with DPPC, DSPC, and DOPC phospholipids, and loaded lincRNA-EPS to prepare a novel “leukosome” for alleviating inflammation injuries of the central nervous system [108]. Liu et al. employed a similar strategy to fuse NK cell-membrane fragments with phospholipids and cholesterol through co-extrusion, thereby constructing a physically modified biomimetic delivery system [109]. By fusing natural thrombocyte membranes with artificial lipid membranes through extrusion, Song et al. formed mixed nanovesicles that retained the targeting ability of thrombocytes as well as the physicochemical properties and drug characteristics of liposomes [111]. Weng et al. fused thrombocyte membrane fragments with soybean phospholipids and cholesterol through co-extrusion; this approach yielded liposome-like vesicles, which extended the in vivo circulation properties of the system [112]. Liu et al. have recently fused erythrocyte membranes with thrombocyte membranes, and thus successfully prepared hybrid membrane nanovesicles loaded with the hypoxia-inducible factor-1α (HIF-1α) inhibitor YC-1 [113].

### 7.2. Chemical Modification

Chemical modification involves chemical reactions that integrate peptides, proteins, lipids, aptamers, or polymers onto the surface of exosomes for brain-targeted delivery, using lipid-binding proteins, membrane-binding proteins, or lipid–lipid interactions. It should be noted that surface protein inactivation may occur during chemical modification [118,119].

By utilizing the sulfonamide click reaction, cell membrane vesicles modified with glioma-targeting peptides can successfully cross the blood–brain barrier after intravenous injection [120]. Thiolation is a commonly used method to modify cysteine residues in proteins, in which the cysteine thiols can be oxidized to form disulfide bonds (-S-S). Xu et al. activated the surface of thrombocyte membranes through sulfhydrylation to generate reactive thiol groups (-SH), which were then chemically conjugated to the thrombolytic drug rt-PA using a bifunctional maleimide linker [76]. Eventually, a surface-loaded drug biomimetic delivery system for brain targeting was developed.

### 7.3. Genetic Modification

Cell membrane functionalization can be obtained by stimulating overexpression of specific membrane protein receptors using exogenous substances, and then constructing biomimetic delivery systems. Shi et al. cocultured Fe_3_O_4_ NPs with mesenchymal stem cells (MSCs) and obtained the cell membranes from MSCs that overexpressed CXCR4, which served as the shell of the NPs [79]. Ma et al. used lentiviral transduction to engineer neural stem cells that overexpressed CXCR4 and isolated their cell membranes to construct a thrombus-targeting biomimetic delivery system [77]. Feng et al. differentiated human promyelocytic leukemia (HL-60) cells to obtain the cell membranes with high expression levels of β2 integrin, LFA-1, and Mac-1, for constructing neutrophil-like nanocarriers [84]. Zhong et al. transfected endothelial cells with a lentiviral vector encoding mouse integrin alpha-4 (Itga4) to obtain engineered endothelial cells with overexpressed VLA-4, which were then used to construct biomimetic delivery systems with the cell membranes as the shell [92].

## 8. Advantages of Cell Membrane-Derived Nanovehicles

### 8.1. Prolonged Circulation

Surface modification with polyethylene glycol (PEG) is a classic “stealth” approach to enhance the in vivo circulation time and bioavailability of NPs. PEG compounds create a protective coating in the body, which prevents drugs from being attacked by metabolic enzymes. They also decrease the binding of drugs to blood proteins, thereby reducing their filtration and clearance from the bloodstream [121,122]. However, repeated administration of PEG-modified liposomes can lead to their rapid clearance from circulation, which is known as accelerated blood clearance (ABC) [123,124,125].

However, CMNVs achieve prolonged circulation by means of cell–cell interactions. CD47, a self-marking transmembrane protein with a molecular weight of 50 kDa, interacts with the inhibitory receptor SIRPα, thereby inhibiting phagocytosis by macrophages [22]. CD47 is widely expressed on the surfaces of different cell types, including erythrocytes, thrombocytes, leukocytes, epithelial and endothelial cells, fibroblasts, mesenchymal cells, and various tumor cells [23]. Consequently, CMNVs possess inherent abilities to evade the immune system and achieve prolonged circulation. For example, erythrocyte membrane-coated NPs contain immunomodulatory proteins such as CD47, CD59, C8 binding protein, and complement factors, which contribute to the immune evasion ability of these NPs. Furthermore, the hydrophilic sugars and negatively charged sialic acid residues on the surface of erythrocyte membranes contribute to the structural formation and long-term circulation properties of biomimetic erythrocyte membrane-coated NPs.

### 8.2. Penetration of BBB

The presence of the blood–brain barrier poses a significant challenge to drug delivery to the brain. The blood–brain barrier tightly regulates the transport of drugs and NPs from the bloodstream to the brain, limiting the passage of macromolecules, small molecules, proteins, antibodies, and nucleic acids. Penetration of the blood–brain barrier involves receptor recognition and intracellular transport. CMNVs can facilitate the penetration of the blood–brain barrier by leveraging natural delivery processes, including energy and nutrient supply, chemotactic effects, or recruitment mechanisms. Similar to the natural entry of immune cells, macrophage nano systems coated with cell membranes can modulate the permeability of endothelial cells by interacting with transmembrane proteins such as integrin α4 and β1. This allows them to cross the blood–brain barrier via the paracellular pathway, ensuring their efficient penetration into the brain. During the processes of ischemia and reperfusion, the disruption of the blood–brain barrier integrity leads to the recruitment and infiltration of macrophages into the ischemic brain within a few hours. Thus, macrophage membrane-coated NPs present a promising strategy for precise delivery of nanocarriers to ischemic lesions. Some malignant tumor cells, such as breast cancer cells, can easily infiltrate the brain through the blood–brain barrier and form metastases. In the context of metastasis, syndecan-1 (CD138) on 4T1 cancer cell membranes can bind to endothelial cell adhesion molecule-1 (CD31) on thrombocytes, endothelial cells, and leukocytes (e.g., monocytes and neutrophils) surrounding the brain vasculature, thereby promoting adhesion and facilitating migration across the blood–brain barrier [24,25,26].

### 8.3. Targeting

Thrombocytes play crucial roles in hemostasis, thrombosis, inflammatory responses, and tumor metastasis [126]. The surface expression of integrins (including αIIb, α2, α5, α6, β1, β3) and other transmembrane proteins (such as GPIbα, GPIV, GPV, GPVI, GPIX, and CLEC-2) on thrombocytes provides thrombocyte membrane-coated particles with significant potential for thrombosis treatment [27]. Overexpression of vascular cell adhesion molecule 1 (VCAM1) on 4T1 cancer cell membranes exhibits high affinity for very late antigen-4 (VLA-4) on lymphocytes, monocytes, and eosinophils. Since thrombocytes and leukocytes preferentially accumulate at sites of brain inflammation, tumor cell membrane biomimetic delivery systems can target ischemic regions in the brain. Based on the observation that 4T1 cancer cells possess the ability to penetrate the blood–brain barrier during brain metastasis, He et al. developed a pH-sensitive polymeric nanocore encapsulated with a 4T1 cell membrane to create CMNVs. This novel construct demonstrates a remarkable capacity for specifically targeting cerebral ischemic lesions, achieving a delivery efficiency that is 4.79 times higher in the ischemic hemisphere compared to the normal hemisphere. Additionally, CMNVs significantly enhance microvascular reperfusion in the ischemic hemisphere, resulting in a substantial 69.9% reduction in infarct volume [94].

During ischemia and reperfusion, intercellular adhesion molecule-1 (ICAM-1), VCAM-1, and P-selectin are overexpressed on endothelial cells [117]. These molecules can interact with their corresponding ligands on the surface of leukocytes (e.g., CD11, CD18, and CD44), leading to the aggregation of leukocytes, particularly macrophages and neutrophils, in the ischemic region [28]. Hence, macrophage membrane-coated NPs demonstrate high targeting efficiency in various inflammatory diseases.

Physical modification of the cell membrane surface can further enhance the targeting ability of the delivery vehicle. SHp [81] and CREKA [80] have been attached to the hydrophilic end of PEG-DSPE and inserted into the outer cell membrane of the biomimetic delivery system to enhance its targeting ability to ischemic sites in the brain, respectively. Wang et al. and Xu et al. successfully modified RGD on the cell membrane of thrombocytes and erythrocytes, respectively, to improve their targeting to the brain sites of thrombosis [75]. Gao et al. modified T807 (AV-1451) on the surface of erythrocyte membrane biomimetic carriers to precisely target AD lesions [78]. Using magnetic Fe_2_O_3_ NPs as the core to prepare biomimetic delivery systems allows for targeted delivery through local exogenous magnetic fields [83].

In addition, engineered cell membranes with highly expressed specific receptors can enhance the targeting properties of biomimetic systems. Engineered MSCs with overexpressed CXCR4 [77,79] were developed into CMNVs to enhance the targeting ability to the injured brain and effectively target the CXCR4–CXCL12 axis. Human promyelocytic leukemia (HL-60) cells with high expression of β2 integrins, LFA-1, and Mac-1 were used to construct CMNVs, which exhibited superior targeting to inflammatory brain microvascular endothelial cells [84]. Zhong et al. fabricated carriers using engineered endothelial cell membranes with overexpressed VLA-4, thereby improving the selectivity and targeting toward vascular inflammatory cells [92].

### 8.4. Controlled Release Delivery

It is widely acknowledged that reperfusion after ischemia leads to the upregulation of a significant amount of toxic ROS, resulting in neuronal damage. The high levels of ROS in the ischemic region can be utilized as intelligent and sensitive triggers for controlling drug release, which can be applied in the development of site-specific drug delivery systems. In CMNVs with a core-shell structure, redox-sensitive features were incorporated into the nanocore through the preparation of boronic esters-modified dextran [81], phenylboronic acid propylene glycol ester-modified chitosan oligosaccharide [127], and poly(vanillin aldehyde-citric acid) (PVAX) [128]. These modifications allow for ROS-responsive drug delivery. In another study [79], dopamine molecules were polymerized to form polydopamine nanocores, which were combined with Zn^2+^ ions to form complexes. The oxidation of polydopamine nanocores and the disruption of the complex structure under high concentrations of hydrogen peroxide resulted in the rapid release of the loaded drug. Additionally, a prednisolone-polymer complex has been developed using ROS-sensitive linkers, which was able to disassemble at the site of a thrombus with a high ROS concentration [128]. Modification of the cell membrane structure can also achieve ROS sensitivity. Weng et al. inserted DSEP-SeSe-PEG2000 block copolymer onto the surface of a biomimetic nano delivery system, which enabled ROS-responsive controlled drug release [112].

Notably, pH changes can also be utilized to control drug release, considering the specific pH values of different biological environments such as blood (pH 7.3–7.4), early endosomes (pH 6.3), late endosomes (pH 5.5), and lysosomes (pH 4.7). Poly (ethylene glycol)-poly (diisopropylamino) methacrylate (PEG-PDPA) undergoes structural changes under different pH conditions. In the tumor cell membrane biomimetic delivery system constructed by He et al., PEG-PDPA was used as the nanocore to encapsulate succinobucol (SCB) [94]. The biomimetic delivery system remained stable in the neutral pH of the bloodstream and released SCB in a pH-responsive manner at the target site to exert therapeutic effects. Similarly, Guo et al. modified pH-sensitive phenylboronic acid onto PEI and constructed a thrombus microenvironment-responsive nanocore, which was loaded with tPA [87]. The nanocore was then coated with thrombocyte membranes to enhance its targeting ability to sites of thombosis.

### 8.5. Collaborative Diagnosis and Treatment

Multimodal therapy can be implemented by loading multiple functional molecules onto the surface of the cell membrane and/or nanocore. For example, the combination of a fluorescent probe [91], photosensitizers [105], superparamagnetic iron oxide (SPIO) [129], and drugs can enable precise localization and imaging of the thrombus site. Additionally, the simultaneous loading of combinations such as α-hydroxypropyl sulfonic acid (LA)-cannabidiol (CBD) [130], bortezomib-tPA [97], Dox-heparin [93], and Se-ginsenoside Rg1 [131] in CMNVs allows for a multitarget and multilevel therapy approach.

Functionalized nanocores can also exhibit collaborative therapeutic activity by targeting multiple sites. For instance, the co-loading of manganese dioxide (MnO_2_) NPs [64], iron nanoenzymes [84], cerium nanoenzymes [85], and similar agents can synergistically enhance the therapeutic effects with co-loaded drugs. These nanocores work by alleviating damage in the ischemic region through the regulation of the inflammatory microenvironment, ROS clearance, macrophage polarization, and other mechanisms.

## 9. Application of Cell Membrane-Derived Nanovehicles in the Treatment of Ischemic Stroke

tPA is widely used in the treatment of thrombotic diseases such as IS due to its ability to convert plasminogen to plasmin, which promotes the dissolution of thrombi. However, the effectiveness of tPA is limited by its short half-life (only 4–6 min). To overcome this limitation, incorporating tPA into cell membrane biomimetic delivery systems has been explored as a strategy to prolong its circulation time and enhance its bioavailability [54,76,132]. Vankayala et al. employed a dual stealth strategy by modifying the surface of erythrocyte membranes with a PEG shell and incorporating tPA, thereby further improving the drug’s bioavailability [105]. In another study, Yu et al. developed a biomimetic nano delivery platform comprising thrombocyte membrane-coated vesicles encapsulating melanin NPs (MNP) and tPA [133]. This platform demonstrated multiple functions, including scavenging of ROS, photothermal conversion, and natural targeting of thrombus sites. These PLTs–MNP–tPA vesicles exhibited significant adhesion and stagnation properties on damaged vessels and thrombi. Furthermore, activated thrombocytes can interact with neutrophils, thereby preventing their inflammatory infiltration into ischemic areas during stroke.

Reducing the infiltration of inflammatory neutrophils is a promising therapeutic strategy for acute IS. Tang et al. designed thrombocyte-mimicking NPs loaded with baicalein and SPIO to target the thrombus site [129]. This delivery system effectively reduced neutrophil infiltration and decreased the size of the infarct area. Additionally, the incorporation of SPIO allowed for real-time monitoring of inflammatory neutrophils. Rapamycin has been shown to alleviate inflammation injury by inhibiting monocyte infiltration. Wang et al. designed a biomimetic delivery system in which rapamycin was encapsulated within monocyte membranes. This system competes with circulating inflammatory leukocytes, preventing their infiltration and alleviating reperfusion-induced damage [73]. 

Ischemia reperfusion injury commonly occurs during acute treatment of IS, leading to cell damage and death upon reconstitution of blood flow to the ischemic region. Li et al. developed CMNVs loaded with thrombocyte membrane, L-arginine, and γ-Fe_2_O_3_ magnetic material [83]. These NPs rapidly accumulated at the ischemic injury site, facilitated by thrombus targeting and external magnetic field stimulation, to restore blood circulation through local release of L-arginine. Resolvin D1 (RvD1) and Resolvin D2 (RvD2) protect against ischemia reperfusion injury. Weng et al. [112] developed a thrombocyte biomimetic RvD1 delivery platform that targeted the injured area and facilitated dead cell clearance, specialized pro-resolving mediator (SPM) generation, and blood vessel formation during the repair stage. Dong et al. [56] utilized the natural adhesion of neutrophils and endothelial cells to encapsulate RvD2 within neutrophil membrane-derived vesicles, specifically targeting inflammatory brain endothelial cells via ICAM-1–integrin β2 interaction, thereby protecting the brain from reperfusion injury. Liu et al. designed hybrid nanovesicles comprising erythrocyte membrane, thrombocyte membrane, and the drug YC-1 [113]. In a rat model of middle cerebral artery occlusion/reperfusion (MCAO/R), these nanovesicles effectively alleviated ischemia reperfusion injury, preserved the integrity of the blood–brain barrier, and suppressed the activation of microglia and astrocytes (Figure 5). Yu et al. developed CMNVs, termed tPA/MNP@PM, by encapsulating MNP and tPA within thrombocyte membrane vesicles (PM) [133]. These CMNVs have multiple functions, including thrombus-targeting ability, photothermal conversion performance, and ROS-scavenging property of natural melanin, thereby being a promising approach for the treatment of IS (Figure 6).

ROS production at the site of thrombus formation is closely related to inflammation, and results in cell and tissue damage and adverse effects on the blood vessel wall. Therefore, clearing ROS is an effective strategy in thrombus treatment. Su et al. developed a macrophage membrane-based ROS-responsive nanoparticle system for targeting the inflammatory microenvironment using modern herbal medicine [89]. Fu et al. synthesized functionalized azo-bridged L-ascorbic acid nanocomposites (CZ NCs) by reacting zoledronic acid with cerium ions, which were then enveloped by the thrombocyte membrane to serve as a new carrier for the drug probucol [85]. This biomimetic nanoplatform synergistically regulated ROS levels and inflammation, exhibiting efficient multi-enzyme activity for the synergistic treatment of cerebral thrombus. Zhang et al. designed biomimetic vesicles that comprise liposomes and macrophage membrane through fusion to deliver the therapeutic agent lncRNA-EPS, which exhibited strong blood–brain barrier penetration and inflammation-targeting effects, thereby reducing inflammation-induced cytotoxicity, increasing lncRNA-EPS levels in microglia by 77.9%, and promoting neuronal regeneration [108]. 

Regulating the immune microenvironment at the thrombus site is another effective strategy in thrombus treatment. Li et al. employed macrophage membrane camouflage to load the drug fingolimod (FTY) and manganese dioxide (MnO_2_) NPs, which consumed excessive hydrogen peroxide (H_2_O_2_) and released required oxygen (O_2_) upon decomposition in acidic lysosomes, ultimately reducing oxidative stress damage, promoting polarization of M1-type microglia into M2-type, and enhancing the survival of damaged neurons [64] (Figure 7). Feng et al. reported a neutrophil membrane-coated cyanide iron nanozyme biomimetic system that targeted the ischemic brain by promoting the polarization of microglia toward M2-type, decreasing neutrophil recruitment, reducing neuronal apoptosis, and promoting the proliferation of neural stem cells, neural progenitor cells, and neurons [84]. Guo et al. designed a thrombus microenvironment-responsive thrombocyte membrane biomimetic carrier incorporating tPA and the antioxidative molecule protocatechuic aldehyde (PC) [87]. This carrier adhered to damaged endothelial cells, released PC under acidic conditions, and protected mitochondrial function in cardiomyocytes by eliminating excessive ROS generated during reperfusion. Liu et al. proposed a “nanobuffer” strategy using neutrophil biomimetic PLGA NPs loaded with α-linolenic acid (LA) and cannabidiol (CBD) for targeted delivery in the ischemic brain region. This approach reshaped the ischemic microenvironment by eliminating ROS, inhibiting inflammation, and attenuating oxidative stress [130].

Utilization of engineered cell membrane-derived biomimetic delivery systems offers advantages in the treatment of IS. Neural stem cells (NSCs), a specific type of stem cell with the capacity for self-renewal and multidirectional differentiation, have been investigated for their ability to selectively target the ischemic brain region. NSC targeting is driven by the chemotactic factor CXCR4, which interacts with its ligand SDF-1 in the ischemic microenvironment. Ma et al. developed PLGA NPs wrapped with an NSC membrane, in which CXCR4 was overexpressed through genetic modification [77]. This approach significantly enhanced the delivery efficiency of the anti-edema drug glyburide. Shi et al. utilized CXCR4-mediated chemotaxis to target the ischemic brain by employing engineered CXCR4-overexpressing MSC membranes as the outer shell of the NPs [79]. The nanoparticle core, which comprised PDA, contained A151, a cGAS inhibitor with antioxidant and anti-inflammatory effects. Not only did these CMNVs improve the homing of the nanoparticles to the damaged brain and efficiently target the CXCR4–CXCL12 axis, but they also served as a “nanodecoy” for CXCL12 deletion, thereby reducing the infiltration of peripheral inflammatory cells. In addition, A151 inhibited the cGAS pathway and led to microglia polarization towards an anti-inflammatory M2-like phenotype (Figure 8).

## 10. Challenges

Similar to other nano delivery systems, the ultimate goal of CMNVs is their translation from the laboratory to clinical applications. However, most of the CMNVs-related research for brain diseases is still in the preclinical stage, and their clinical translation needs to overcome several key issues, including the establishment of quality standards and evaluation systems, the stability of large-scale production, the adaptation of analysis and detection techniques, drug-loading capacity, as well as safety and long-term metabolic characteristics in the human body.

A major issue is the structure characterization and quality control. Traditional nano delivery systems have clear and well-defined structural information, which can be analyzed and verified using various techniques such as nuclear magnetic resonance, infrared spectroscopy, mass spectrometry, gel electrophoresis, and high-performance liquid chromatography to ensure stability and reproducibility between batches. However, cell membranes are biomimetic samples derived from living organisms and contain rich biological information. The functionality of cell membranes is closely related to their preparation process, which can affect the structure and functionality of the membranes. Currently, there is a lack of comprehensive evaluation criteria for characterizing cell membrane biomimetic delivery systems. Qualitative and semiquantitative studies of key proteins have been conducted using gel electrophoresis, protein immunoblotting, and proteomic methods [76]. Therefore, further research is needed to standardize and optimize the extraction, purification, and other key steps in the preparation of cell membranes, as well as to improve the techniques for characterizing and evaluating their structure and quality. This is essential to ensure the stability and reproducibility of CMNVs between different batches.

Another challenge is the large-scale production of CMNVs with stability and consistency. Obtaining large quantities of cell membrane samples poses difficulties due to the characteristics of cell culture and amplification. Therefore, there is a need to explore simple and rapid manufacturing techniques, establish standardized and automated production processes, and obtain high-quality, scalable, and reproducible cell membrane samples. Considering the structural similarity between cell membranes and liposomes, cell membrane-lipid hybrid samples can be prepared through membrane hybridization and fusion techniques [107]. This approach preserves the biomimetic characteristics of cell membranes and expands the production scale of the samples, providing new ideas and directions for large-scale production.

For CMNVs, not only should an ideal drug-loading method have high encapsulation efficiency and cargo capacity, but it should also maintain the integrity of the membrane structure and function. Therefore, the main challenge for these systems is to effectively load therapeutic drugs without compromising the integrity of the membranes. For shell-core NPs, traditional nanocores such as PLGA NPs [73,74,75], BSA NPs [78], and dextran NPs [80,81,82] have limited drug-loading capacity. Therefore, covalent linkage between drugs and polymers, followed by self-assembly to form nanocores, can increase drug-loading ability. Additionally, stable loading can be achieved by directly conjugating drugs to membrane proteins through chemical reactions [118,119]. However, exposure to high temperature, pressure, or organic solvents during the process may cause membrane rupture and protein denaturation. The emerging technique of electroporation provides a new means for direct drug loading [101]. By adding drugs and cell membrane vesicles to a conductive solution and applying a transient electric field, transient disruption of the membrane structure occurs. This leads to the formation of small pores through which drugs can enter the vesicles, thereby achieving efficient loading. However, some studies have indicated that electroporation may cause extensive aggregation of nano systems, which hinders their subsequent applications.

Finally, thorough exploration of the safety and long-term metabolic processes of CMNVs in humans is imperative. Apart from their target sites, CMNVs tend to aggregate in other organs, particularly the liver. The influence of the “protein corona” formed during CMNVs circulation on their targeting efficiency and efficacy should also be taken into consideration. Hence, a comprehensive evaluation of the long-term safety and potential adverse effects regarding the absorption, distribution, and metabolism of CMNVs in the human body must be conducted prior to their clinical translation. Furthermore, the majority of CMNVs efficacy validation is based on animal disease models, such as rats, rabbits, or mice. However, these models fail to fully replicate the conditions experienced by human patients in clinical practice. Additionally, most patients with brain diseases are affected by multiple underlying disorders. Consequently, it is crucial to establish more appropriate animal models that can simulate the realistic circumstances encountered by humans.

## 11. Prospects

Looking to the future, CMNVs have emerged as a promising strategy for the development of multifunctional and diverse NPs for treating diseases of the central nervous system. By utilizing the surface proteins retained on the biomimetic membrane, CMNVs possess advantageous properties such as immune evasion, specific targeting, low toxicity, and prolonged circulation, which make them an appealing option for addressing challenges in the treatment of IS. Several biomimetic delivery systems utilizing CMNVs have already been developed, including erythrocyte, thrombocyte, macrophage, neutrophil, and stem cell mimics. Moving forward, CMNVs will continue to play an increasingly important role in treating diseases of the central nervous system.

The integration of diagnostic and therapeutic CMNVs allows for real-time monitoring and noninvasive visualization of therapeutic effects. Furthermore, the biocompatibility of CMNVs can effectively protect diagnostic agents from rapid clearance by the mononuclear phagocyte system. Previous studies have investigated the integration of photosensitizers, fluorescent dyes, and other diagnostic and therapeutic agents into CMNVs for early diagnosis and intervention in IS. With further advancements in imaging techniques, CMNVs have the potential to revolutionize diagnosis and therapy integration for brain diseases.

In the field of tumor therapy, combinations such as chemotherapy–immunotherapy, chemotherapy–photodynamic therapy, and chemotherapy–chemodynamic therapy have considerably enriched strategies for tumor treatment. Similarly, combining traditional drug treatment modalities with hyperthermia, oxygen supply, and sonodynamic coagulation therapy through targeted design and functionalization can further enhance the efficacy of IS treatment [134,135,136].

CMNVs present new opportunities for targeted delivery of bioactive macromolecules to the brain. One example is the modification of heparin’s structure and the construction of nanocores, which allows for efficient delivery of heparin (1134 Da) across the blood–brain barrier. Su et al. employed a similar strategy by linking Angelica polysaccharide and ethyl ferulate via an oxalate bond to form a nanocore [89]. By coating the nanocore with macrophage membranes, they achieved successful delivery of the active polysaccharide to the brain. Thus, it is possible to develop active macromolecules as nanocores using methods such as nanoprecipitation, emulsification, or self-assembly, and then encapsulate them with cell membranes to construct CMNVs for brain-targeted delivery.

CMNVs demonstrate capabilities for both passive and active targeting of ischemic regions. This targeting is primarily driven by inflammatory chemotaxis and ligand binding. However, CMNVs can sometimes suffer from inadequate drug release within the lesion area. While stimuli-responsive nanoplatforms have been developed, the distinctive core-shell structure of CMNVs may affect drug release. Therefore, it is essential for future research to investigate innovative approaches, such as combining thermal-sensitive liposomes with cell membranes. This combination would allow for controlled drug release at the target site through external temperature stimulation.

Bacterial-derived outer membrane vesicles (OMVs) are specifically recognized by Toll-like receptors (TLRs) on neutrophils, promoting neutrophil internalization of the vesicles. This suggests that OMVs have the potential to serve as neutrophil-mediated brain-targeted nanoplatforms. Recently, researchers have utilized neutrophil-attached bacterial-derived OMVs to enhance the delivery of pioglitazone (PGZ) in the treatment of IS [137]. By encapsulating PGZ within OMVs, the resulting OMV@PGZ NPs retained the functions associated with bacterial outer membranes, which made them an ideal target for neutrophil uptake. The study demonstrated that OMV@PGZ simultaneously inhibited the activation of NLRP3 inflammasomes and ferroptosis, thereby alleviating reperfusion injury and exerting neuroprotective effects. The exploration of bacterial membrane-derived biomimetic delivery systems presents a new research direction for the design of future CMNVs.

Finally, considering that CMNVs are an emerging nano delivery system, their biocompatibility requires further evaluation. Factors such as nanoparticle size, shape, and surface properties play significant roles in determining their biocompatibility. For cell membrane biomimetic delivery systems, especially those derived from tumor cells, comprehensive evaluation of biocompatibility is necessary to ensure their sustainable development in medical and biotechnological applications.

## Figures and Tables

**Figure 1 pharmaceutics-16-00006-f001:**
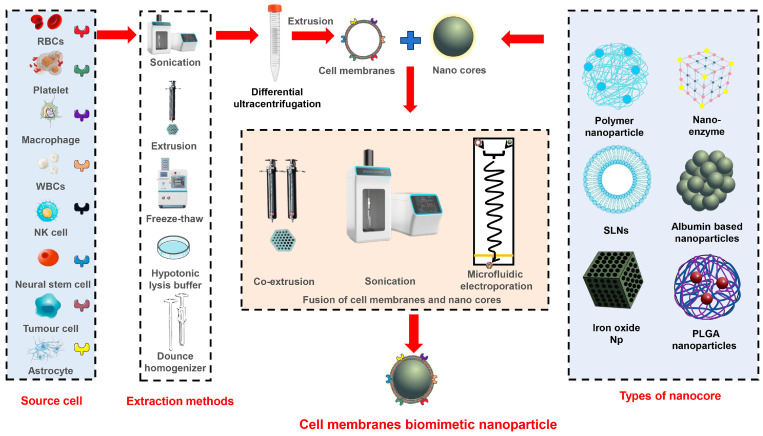
Schematic representation of different methods for the construction of core-shell structured CMNVs.

**Figure 2 pharmaceutics-16-00006-f002:**
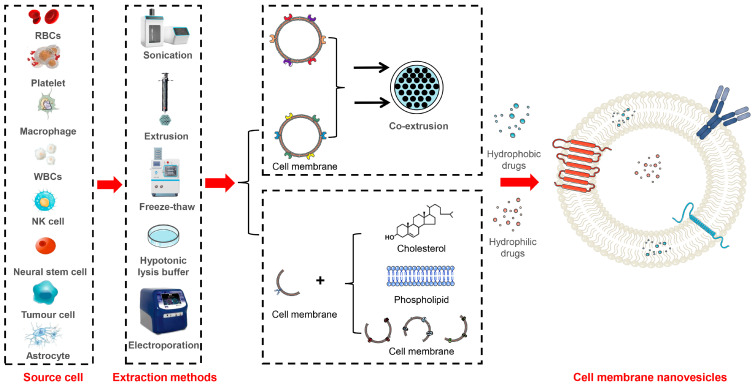
Schematic representation of different methods for the construction of cell membrane nanovesicles.

**Figure 3 pharmaceutics-16-00006-f003:**
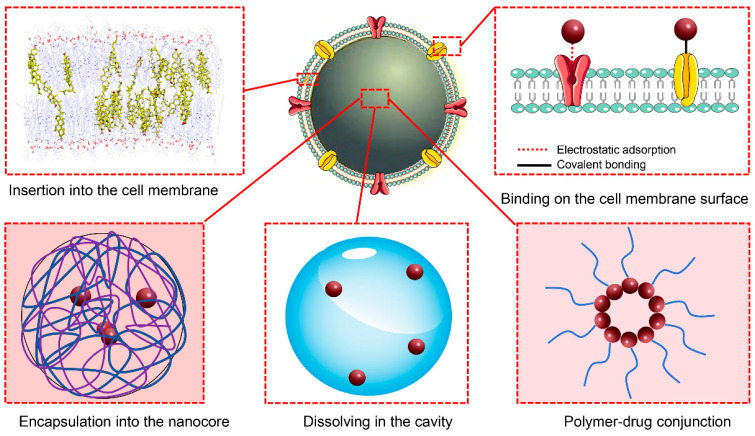
Schematic representation of the drug-loading modes.

**Figure 4 pharmaceutics-16-00006-f004:**
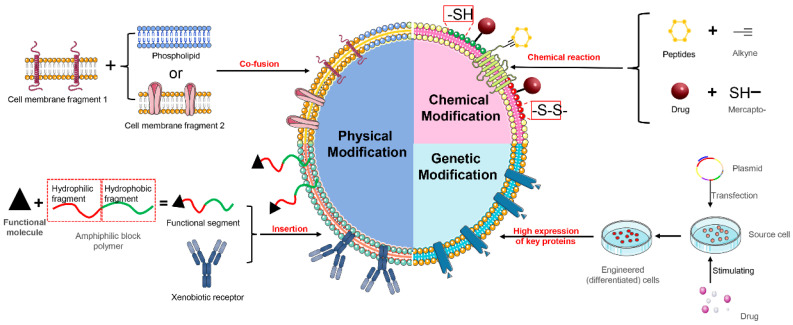
Schematic representation of surface modification of cell membranes.

**Figure 5 pharmaceutics-16-00006-f005:**
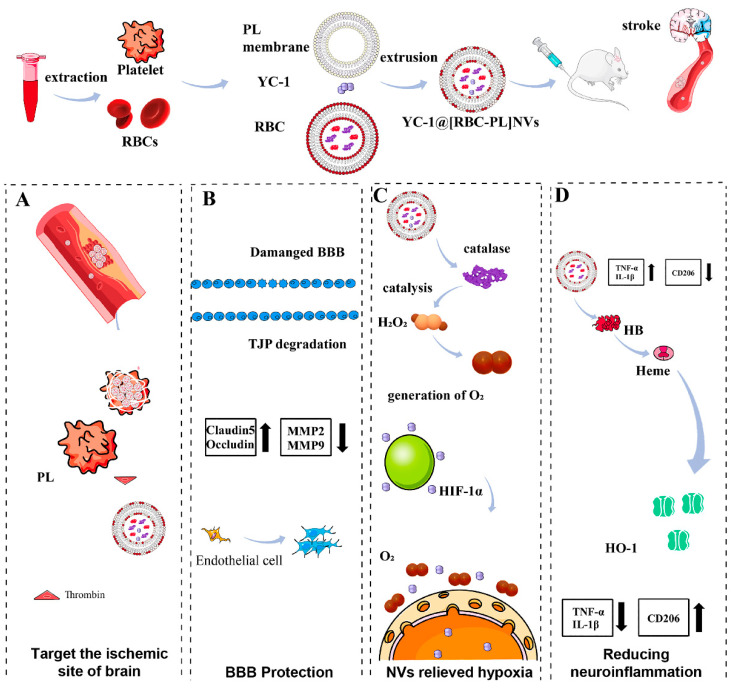
(**A**): The hybrid membrane nanovesicles (YC-1@[RBC-PL] NVs) for the treatment of ischemic stroke (IS). The constructed nanovesicles comprised a combination of erythrocyte (RBC) membrane and thrombocyte (PL) membrane, with the HIF-1α inhibitor YC-1 encapsulated within. Upon administration, YC-1@[RBC-PL] NVs adhered to and accumulated in the damaged cerebral vessels within the IS region. (**B**): The NVs protected the breakdown of the blood–brain barrier and alleviated hypoxia in the affected area. (**C**): The interior of the NVs contained catalase, which acted as a catalyst for converting H_2_O_2_ into O_2_. (**D**): YC-1 reduced the expression of HIF-1α, whereas the presence of heme oxygenase-1 (HO-1) led to the upregulation of HIF-1α expression. Consequently, this modulation effectively mitigated neuroinflammation. Reprinted from Liu et al. (2023), with permission from Elsevier [113].

**Figure 6 pharmaceutics-16-00006-f006:**
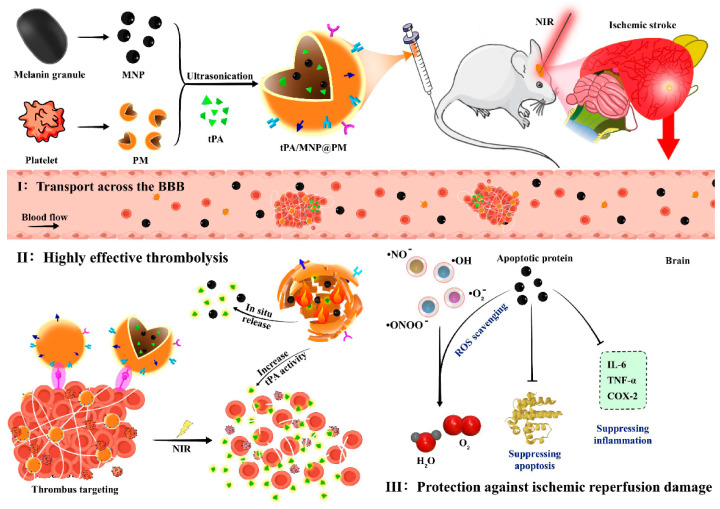
The tPA/MNP@PM system for the treatment of ischemic stroke (IS) and reperfusion injury. In this system, thrombocyte membrane vesicles (PMs) were loaded with melanin nanoparticles (MNPs) derived from natural sources and tissue plasminogen activator (tPA) to create the tPA/MNP@PM (tMP) formulation. The presence of PM on tMP allows for stealth behavior and targeted delivery to thrombus sites. Additionally, the photothermal properties of MNP enable the release of tPA from tMP within the thrombus upon localized near-infrared (NIR) irradiation, facilitated by photothermal-mediated membrane rupture. The resulting local increase in temperature enhances the thrombolytic activity of tPA, leading to accelerated thrombolysis. Following thrombolysis and restoration of blood flow, the released MNP-4.5 can penetrate the blood–brain barrier and migrate to the lesion site. This enables the removal of free radicals generated by oxygen bursts and inhibition of inflammation, thereby preventing reperfusion injury. Reprinted from Yu et al. (2021), with permission from Elsevier [133].

**Figure 7 pharmaceutics-16-00006-f007:**
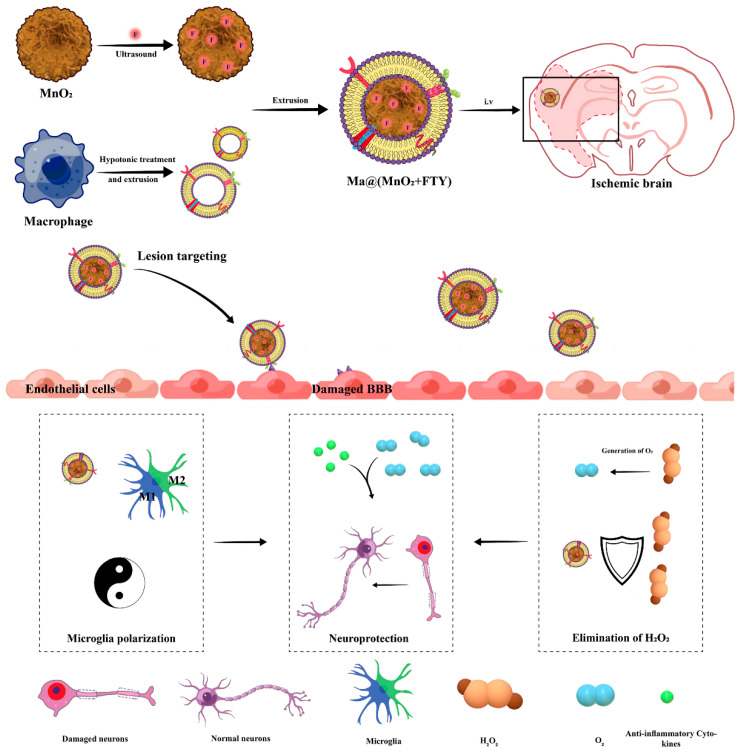
Ma@(MnO_2_+FTY) for the treatment of ischemic penumbra. The nanoparticles are coated with macrophage membrane, which grants them inflammation-targeting chemotactic abilities. The presence of MnO_2_ nanoparticles, known for their large surface area, enables effective scavenging of excessive reactive oxygen species (ROS) and the generation of oxygen (O_2_) to rescue compromised neurons. This ROS consumption process additionally inhibits the NF-κB signaling pathway in microglia, leading to a reduction in the proinflammatory response. FTY, housed within MnO_2_ nanoparticles via electrostatic interaction, facilitates the phenotypic transformation of microglia, ultimately reversing the proinflammatory microenvironment. This transition is achieved by activating the signal transducer and activator of the transcription 3 (STAT3) pathway. Reprinted from Li et al. (2021), with permission from Wiley [64].

**Figure 8 pharmaceutics-16-00006-f008:**
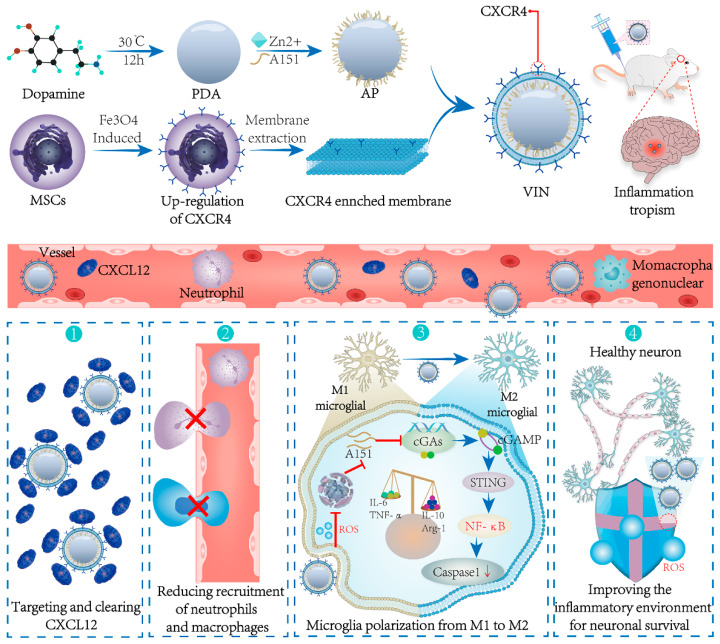
The engineering CXCL12 biomimetic decoy-integrated versatile immunosuppressive nanoparticle (VIN) for the management of overactivated brain immune microenvironment. The shell of the VIN serves two key functions, namely, enhancing the homing of the nanoparticles to cerebral ischemic lesions and effectively adsorbing and neutralizing CXCL12. In that way, infiltration of peripheral neutrophils and mononuclear macrophages is blocked. Additionally, the VIN is loaded with A151, which can inhibit the cGAS–STING pathway. This inhibition leads to the polarization of microglia towards an anti-inflammatory M2-like phenotype. As a result, the VIN demonstrates remarkable therapeutic effects in terms of reducing mortality rates, minimizing infarct volume, and protecting the neurogenic functions of neurons. Reprinted from Shi et al. (2022), with permission from Wiley [79].

**Table 1 pharmaceutics-16-00006-t001:** CMNVs with a core-shell structure for the treatment of IS.

Cell Membrane	Surface Modification	Nanocore	Nanocore Preparation Method	Fusion Method	Drug	Loading Profile	Remark	Ref.
Macrophage membrane (MM)	-	MnO_2_	Ultrasonication	Extrusion	Fingolimod (FTY)	Adsorption	pH-sensitive	[64]
Monocyte membrane	CD47, integrin α4, and integrin β1	PLGA	Nanoprecipitation	Extrusion	Rapamycin (RAP)	Encapsulation	-	[73]
Erythrocyte membrane	-	PLGA	Dialysis	Extrusion	Rapamycin (RAP) and atorvastatin calcium (AC)	Co-encapsulation	-	[74]
Thrombocyte membrane	DSPE-PEG2000-RGD	PLGA	Double emulsion process	Sonication	Fat extract (FE)	Encapsulation	-	[75]
Thrombocyte membrane	-SH	PLGA	Nanoprecipitation	Sonication	(rt-PA)	Adsorption	-	[76]
Nerve stem cell membrane	Overexpression of CXCR4	PLGA	O/W emulsion	Extrusion	Glyburide	Encapsulation	-	[77]
Erythrocyte membrane	DSPE-PEG3400-T807 and DSPE-PEG2000-TPP	HSA	Emulsification ultrasonication	Extrusion	Curcumin (CUR)	Encapsulation	-	[78]
Mesenchymal stem cell (MSC) membrane	Overexpression of CXCR4 and introduction of Zn^2+^	PDA		Sonication	A151	Adsorption	H_2_O_2_-sensitive	[79]
Erythrocyte membrane	DSEP-PEG-CREKA	Dextran	Self-assembly	Extrusion	Tirofiban	Covalent bonding	-	[80]
Erythrocyte membrane	SHp-PEG-DSPE	PHB-dextran	Assembly	Co-sonication	NR2B9C	Encapsulation	ROS-sensitive	[81]
Macrophage membrane (MM)	F4/80, CD47, and integrin α4/β1	Dextran-g-PBMEO	Self-assembly	Extrusion	Kaempferol (KPF)	Encapsulation	-	[82]
Thrombocyte membrane	α6β1, α5β1, β2β1, avβ3, and αIIbβ3, glycoproteins	γ- Fe_2_O_3_	One-pot protocol	Extrusion	L-arginine	Co-encapsulation	-	[83]
	GPVI and GPIb-IX-V, CD59, CD55, CD47, CD31							
Neutrophil-like cell membrane (NCM)	High expression of β2 integrin, LEA-1, Mac-1	PVP	-	Extrusion	Fe [(CN_6_)]^3−^	Cross-linking	-	[84]
Thrombocyte membrane		CZ	Co-assembly	Sonication	Probucol (PB)	Co-assembly	-	[85]
Macrophage membrane (MM)	RVG29-PEG-DSPE	SLNs	Solvent injection	Extrusion	Genistein (GS)	Loading	-	[86]
	TPP-PEG-DSPE							
platelet membrane	PEI	PBA		Extrusion	tPA, PC	Loading	slightly acid condition	[87]
erythrocyte membrane	S-S	PLL-SH		self-assembly	Heparin (Hep)	Electrostatic attraction		[88]
macrophage membrane		AOE@TMP	Extrusion	Extrusion	Angelica polysaccharide (APS)	Linking		[89]
Thrombocyte membrane	Zn^2+^	MOF	One-pot protocol	Co-extrusion	siRNA	Cross-linking	pH-sensitive	[90]
Erythrocyte membrane	-	PMMP	Self-assembly		LFP	Encapsulation	-	[91]
					Prednisolone (Pred)			
Endothelial cell membrane	Overexpression of VLA-4 and CD47	PEG	Self-assembly	Extrusion	Rapamycin (RAP)	Linking	-	[92]
Thrombocyte membrane	Overexpression of CD62P and GPVI	Succinylated heparin	Self-assembly	Co-sonication	Doxorubicin (DOX)	Self-assembly	-	[93]
4T1 cell membrane	Overexpression of CD47	PEG-PDPA	Extrusion	Extrusion	SCB	Loading	pH-sensitive	[94]

**Table 2 pharmaceutics-16-00006-t002:** CMNVs with a nanovesicle structure for the treatment of IS.

Cell Membrane	Membrane Hybridization	Surface Modification	Fusion Method	Drug	Loading Profile	Ref.
Macrophage	-	-	Ultracentrifugation	Curcumin (CUR)	Phagocytosis	[104]
Erythrocyte membrane	-	DSEP-PEG-CHO	Centrifugation	Tissue plasminogen		[105]
				activator (tPA)	Linking	
				Indocyanine green (ICG)	Encapsulation	
Macrophage membrane (MM)	-	-	Extrusion, centrifugation	Curcumin (CUR)	1. Freeze-thaw cycle. 2. Ultrasonic treatment. 3. Incubation	[106]
Mesenchymal stem cell (MSC) membrane	Fusion with phospholipids	CD47 and some other MSC cell surface proteins and ligands	Sonication	Curcumin (CUR)	Encapsulation	[107]
Macrophage membrane (MM)	Fusion with DPPC, DSPC, and DOPC phospholipids	-	Film dispersion	LincRNA-EPS	Encapsulation	[108]
NK cell membrane	Fusion with phospholipids and cholesterol		Thin-layer evaporation method	Curcumin (CUR)	Encapsulation	[109]
Macrophage membrane (MM)	Ginsenoside Rg5 instead of cholesterol molecules embedded in the phospholipid bilayer	-	-	Paclitaxel (PTX)	Embedment	[110]
Thrombocyte membrane	Fusion with artificial lipid membrane	-	Extrusion	Rapamycin (RAPA)	Encapsulation	[111]
Thrombocyte membrane	Fusion with soybean lecithin and cholesterol	-	Extrusion	Resolvin D1 (RvD1)	Encapsulation	[112]
Erythrocyte membrane	Fusion with thrombocyte membrane	-	Extrusion	YC-1	Encapsulation	[113]

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
