# Peer review of "Cell Membrane-Derived Nanovehicles for Targeted Therapy of Ischemic Stroke: From Construction to Application"

_pharmaceutics, 2023, doi:10.3390/pharmaceutics16010006_

Round 1

Reviewer 1 Report

Comments and Suggestions for Authors

The present work is a review article regarding Cell membrane–derived nanovehicles for targeted therapy of ischemic stroke: From construction to application. Because of the barely explored nature of the subject, the study is of interest but needs some amendments. Please consider the following observations: 

  1. On page 3, line 107. Change the heading subsection for Eryhrocytes. 
  2. Some references do not correspond. Please review them carefully and cite them properly.
  3. Please write the formulas adequately. Use sub-indices and super-indices as required.
  4. Please expand the contents of subsections 3.1 to 3.5.
  5. The references in Table 1 do not correspond. Please review the references in the whole manuscript.
  6. The sub-sections 4.1 and 4.1.1 do not contain references. Please cite the information in the sections. 
  7. Please expand the Albumin sub-section. 
  8. The sub-section 4.1.2 has the heading of metal nanoparticles. The examples in this section correspond to metal oxide nanoparticles. Please label precisely. 
  9. On page 11 and line 323, the authors refer to particle size using voltage units. Please explain or correct. 
  10. On page 15, in heading 7.2. Please review the heading. Is chemical modification?

Comments on the Quality of English Language

Minor revision

Author Response

Reviewer 1

The present work is a review article regarding Cell membrane–derived nanovehicles for targeted therapy of ischemic stroke: From construction to application. Because of the barely explored nature of the subject, the study is of interest but needs some amendments. Please consider the following observations: 

  1. On page 3, line 107. Change the heading subsection for Eryhrocytes. 

Response: Revised, as seen in page 3, line 135.

  1. Some references do not correspond. Please review them carefully and cite them properly.

Response: We thank the kind comment. We have carefully checked the references, and cited them properly.

  1. Please write the formulas adequately. Use sub-indices and super-indices as required.

Response: We thank the kind comment and use sub-indices and super-indices as required.

  1. Please expand the contents of subsections 3.1 to 3.5.

Response: We are grateful for the constructive suggestion. As per your recommendation, we have expanded the content of sections 3.1 to 3.5 and included relevant citations from recent literature examples. These can be found on pages 3 to 5, lines 141 to 204 (highlighted in red). 

  1. The references in Table 1 do not correspond. Please review the references in the whole manuscript.

Response: We have carefully reviewed Table 1 and made the necessary updates to ensure that the references are properly cited.

  1. The sub-sections 4.1 and 4.1.1 do not contain references. Please cite the information in the sections. 

Response: we thank the kind suggestion. Indeed, the detailed descriptions of different nanocores such as PLGA nanoparticles, HSA nanoparticles, PDA nanoparticles, etc., are all included under section 4.11. Each description is accompanied by relevant literature citations.

  1. Please expand the Albumin sub-section. 

Response: we have expanded the albumin section, as seen in page 8 to 9, line 288 to 298.

  1. The sub-section 4.1.2 has the heading of metal nanoparticles. The examples in this section correspond to metal oxide nanoparticles. Please label precisely. 

Response: We are grateful for the thoughtful comment. We have revised the heading of section 4.1.2 to "Metal Oxide Nanoparticles".

  1. On page 11 and line 323, the authors refer to particle size using voltage units. Please explain or correct. 

Response: we have corrected the errors, as seen in page 11, line 419.

  1. On page 15, in heading 7.2. Please review the heading. Is chemical modification?

Response: we have corrected the errors, as seen in page 15, line 521.

Reviewer 2 Report

Comments and Suggestions for Authors

1.      In Section 3., the cellular structure and function as of NK cells should be added.

2.      Reframe the abstract and update it with new keywords related to manuscript content.

3.      Several nanoparticle-based systems are formulated with the help of cell membranes including cancer cells and stem cells. My suggestion is to add some latest research reports based on cancer cells or stem cell-based nanoparticle systems.

4.      Why cell membranes for nanoparticle development? The rational uses of cell membranes for nanoparticles should be added to the introduction section.

5.      In section 4. Some of the paragraphs are too short. Elaborate it with proper references. Just providing the basic information’s are not sufficient.

6.      Importance of cell membrane i.e., cell-cell interaction should be emphasized.

Comments on the Quality of English Language

Moderate English editing is required. 

Author Response

Reviewer 2

1.In Section 3., the cellular structure and function as of NK cells should be added.

Response: We thank this kind suggestion. We have revised this section, as seen in page 4, line 177 to 189.

  1. Reframe the abstract and update it with new keywords related to manuscript content.

Response: we have rewritten the abstract and update the new keywords, as seen in page 1, line 11 to 23.

  1. Several nanoparticle-based systems are formulated with the help of cell membranes including cancer cells and stem cells. My suggestion is to add some latest research reports based on cancer cells or stem cell-based nanoparticle systems.

Response: We thank this kind suggestion. Accordingly, we have added the latest research reports about cancer cells/stem cell-based nanoparticle systems for IS treatments, as seen in page 5, line 216 to 229, page 17, line 600 to 607.

  1. Why cell membranes for nanoparticle development? The rational uses of cell membranes for nanoparticles should be added to the introduction section.

Response: We appreciate this helpful suggestion. The introduction section now includes the appropriate discussion on the rational uses of CMNVs, which can be found on page 2, lines 44 to 81 (highlighted in red).

  1. In section 4. Some of the paragraphs are too short. Elaborate it with proper references. Just providing the basic information’s are not sufficient.

Response: We would like to express our gratitude for this kind comment. As per your suggestion, we have revised the section accordingly. Several paragraphs have been expanded, and you can find the revised content on pages 8 to 10 (highlighted in red).

  1. Importance of cell membrane i.e., cell-cell interaction should be emphasized.

Response: We would like to express our gratitude for this kind comment. In response to your feedback, we have placed greater emphasis on the cell-cell interaction of the CMNVs. Specifically, this information can now be found on page 2, lines 66 to 81.

Reviewer 3 Report

Comments and Suggestions for Authors

The article from Cui et al. Is an extensive and detailed review on cell-derived nanoparticle-based drug delivery in the specific context of ischemic stroke treatment. The review is detailed and instructive.

The authors deal with pharmaceuticals aspects such as preparation, surface modification, drug loading. A consistent number of references are given for each chapter so that the reader has a comprehensive view of the scientific domain.

Globally the manuscript is well written and easy to follow.

A point worth further development is related to clinical aspects: are there cell-derived nanoparticles used in clinics or make the subject of FDA approval? What about clinical trials?

Comments on the Quality of English Language

Quality of English is good, some typos to be corrected before publication.

Author Response

Reviewer 3

The article from Cui et al. Is an extensive and detailed review on cell-derived nanoparticle-based drug delivery in the specific context of ischemic stroke treatment. The review is detailed and instructive.

The authors deal with pharmaceuticals aspects such as preparation, surface modification, drug loading. A consistent number of references are given for each chapter so that the reader has a comprehensive view of the scientific domain.

Globally the manuscript is well written and easy to follow.

A point worth further development is related to clinical aspects: are there cell-derived nanoparticles used in clinics or make the subject of FDA approval? What about clinical trials?

Response: We would like to express our gratitude for this kind comment. In response to your feedback, we have revised section 10, the clinical aspects of CMNVs have been emphasized. The revised parts can be found in page 24, line 825 to 831, and page 25, line 874 to 886.

Round 2

Reviewer 1 Report

Comments and Suggestions for Authors

There are still some mistakes that the authors must correct (i.e., section 3.4, Natural Killers). Please review the contents carefully and correct them. Provide a clean version.

Comments on the Quality of English Language

Minor changes required.

Author Response

We have carefully reviewed the manuscript and made revisions to correct the errors, as seen in line 176 and 187 (marked with red)

Reviewer 2 Report

Comments and Suggestions for Authors

The authors revised the manuscript well and can be accepted for publication in its present form. 

Author Response

We carefully reviewed the manuscript and made revisions to correct the errors.